# Ontological Method for the Modeling and Management of Building Component Construction Process Information

Lu Jia [1,*], Yanfeng Jin [1], Yang Liu [1] and Jing Lv [2]

1   School of Infrastructure Engineering, Nanchang University, Nanchang 330047, China;
jinyanfeng@email.ncu.edu.cn (Y.J.)
2   Zhongmei Engineering Grouping, Nanchang 330001, China
*   Correspondence: jialuuser@163.com; Tel.: +86-189-7088-7890

**Abstract:** Knowledge of the construction process plays a decisive role in guiding construction and thus affects the quality of buildings. However, the integration and relevance of information in the traditional process file are poor, and it is easy to ignore the process changes caused by the differences when applying information directly. Therefore, this paper introduces an ontological method for the information modeling and management of the construction process. The proposed method uses machine readable language to integrate process knowledge in a structured way, thereby improving the relevance of information and promoting the reuse, sharing and retrieval of knowledge. At the same time, the semantic web rule language (SWRL) rules are used to model the relevant laws and regulations with reference to the construction product quality acceptance regulations and to correlate the construction process information so as to ensure the compliance of building products and the accuracy of process information. Finally, the feasibility of the method is verified via specific use cases on an ontology implementation platform.

**Keywords:** construction process; ontology; process management; semantic modeling

## 1. Introduction

Quality control is an important part of the production process for building components. The biggest impact on the quality of building components occurs during the construction stage. Quality control in the construction stage is mainly the quality control of the process, which is the basis for achieving the quality goal of the building components. Technical disclosure is the last pass in the pre-control of the process quality. Technical disclosure usually needs to be supported by process files, which are documents that describe the sequence of process activities and related information. Therefore, the preparation of the construction process file is particularly important although, at this stage, the preparation method is generally used to simply apply the construction process files concerning similar building components in the past [1]. Due to the poor degree of informatization in the construction field, the process file has always taken the form of paper or electronic documents. The information contained in these is poor in terms of integration and relevance, and the information expression is not systematic. The reuse mode of simple replication also requires human analysis of the differences in features between building components and the manual modification of process information.

To this end, we study how to present the existing construction process knowledge in a more structured way, improving the correlation between the information, especially be-tween the various processes of the construction activity and the remaining parts, such as the key measures and the resources required. In addition, different structural requirements and construction processes are required for building components with different features, or due to the necessary constraints of the construction parameters under different conditions, which are dynamic. The relevant constraints are usually derived from the building codes

in the field, or some are even stored as the tacit knowledge of experts. People who are not familiar with the provisions of the specification or do not have enough experience are prone to make mistakes in the implementation of relevant decisions. Thus, how to support the automatic inference of process information to guide construction based on constrained knowledge from specifications or experience becomes one of our starting points.

Based on the above research purposes, we propose an overall semantic framework for ontology-based construction process information modeling and application. On the one hand, construction process knowledge is structured by constructing an ontological model, which ensures the correlation between information and facilitates the reuse of knowledge. On the other hand, the relevant building specifications or expert experience and knowledge are modeled as semantic web rule language (SWRL) rules to form a knowledge base. This not only facilitates the sharing of experience and knowledge from engineering practice or experts but also combines with the ontological model to perform the reasoning for the SWRL rules, which provides computer support for the compliance guarantee of the building components and the reasoning behind the process information.

This paper is organized as follows. Firstly, the key technical background of the ontological method is introduced. Then, research concerning the ontological method in the field of building knowledge representation, especially process expression, and research concerning the modeling of normative or empirical provisions as rules to ensure compliance in the field, is reviewed. Subsequently, the paper introduces the development and application methods of the semantic framework construction based on the ontological model. Then, the feasibility of the approach is verified using an ontology implementation platform. Finally, the contribution of the current work and future expectations of the research are described.

## 2. Related Work

### 2.1. Semantic Web and Ontology

The Semantic Web is the father of the World Wide Web Tim Berners-Lee [2] proposed. It is committed to making text information on the Web have the semantics that a computer system can understand. In order to make the disordered knowledge orderly, the Semantic Web provides a set of representation languages and tools designed to describe information, which are used to formally describe the concepts, terms and relationships in a knowledge domain. The development of the Semantic Web provides a solution for the structured organization and reuse of domain-specific knowledge.

Ontology originated in the field of philosophy, and it is mainly used to describe the abstract nature of objective things. It is a philosophical theory that explores objective things and their essences and laws [3]. Neches et al. defined ontology as "the definition of the basic terms and relations that make up the vocabulary of the relevant domain, and the definition of the rules that use these terms and relations to specify the extension of these terms" [4]. With the rapid development of modern information technology and artificial intelligence, ontology is widely used in various fields to realize the reuse and sharing of domain knowledge [5]. As a semantic basis for communication between different subjects within the domain, it is generally considered to contain four meanings: conceptualization, explicit, formalization and sharing [6]. Conceptualization refers to the abstraction of the relevant concepts of the objective world; explicit refers to clarifying the constraint relationship between the concepts or knowledge used; formalization is the setting and coding of concepts or knowledge through specific forms so that computers or humans can recognize and understand their use; and sharing refers to a set of concepts commonly accepted in a specific field.

Ontology is often used to create conceptual models describing information in different fields, which has very strong expressive ability and logical reasoning support. There are many ways to classify ontology, but the essence and laws of these ways are interlinked. They all contain five basic elements: concept (class), relationship, function, axiom and

instance in order to accurately describe knowledge relationships [7]. The mathematical expression can be expressed as:

$$O = \{C, R, F, A, I\} \tag{1}$$

- $C$ can be expressed as a Class, or can also be expressed as Concepts;
- $R$ represents Relations and is the interaction between Classes in the domain;
- $F$ represents Functions and is a specific form of expression to express relations;
- $A$ represents Axioms and refers to the factual description of theorems, rules and other facts in the field of ontology application;
- $I$ represents Instances and refers to the actual object of the Class.

Ontology should be implemented according to the defined ontology description language. With the evolution of the Web, several web-based ontology representation languages have emerged. W3C has endorsed resource description framework (RDF), DAML+OIL, and web ontology language (OWL) as standard ontology-description languages. RDF is a language for describing relationships between elements, represented as a triplet set in a human-machine readable file processable by machines. Numerous existing languages have been derived from the RDF. In this study, we adopted OWL as the ontology coding language. This has become the latest W3C-recommended standard for ontology representation [8]. As an extension of RDF, OWL can provide more primitives to support richer semantic expressions, enabling fast and flexible data modeling and efficient automatic reasoning capabilities.

*2.2. Process Knowledge Model*

Numerous ontologies have been developed to facilitate knowledge management and information extraction across various domains [9].

The ontological model utilized in this study to integrate process information has been extensively employed in the manufacturing and processing domains. These models have reached a relatively advanced level of process knowledge management, including processing sequences, methods, and parameter points [10]. Introducing ontology for product knowledge and information modeling aims to enhance product design and enable the reuse of manufacturing knowledge during the manufacturing process. The core reference body, CROS, can formalize fundamental knowledge related to the steelmaking process and resources [11], facilitating the integration of decentralized data and information to promote knowledge sharing and information management in steelmaking [12]. By employing an ontology-based method to manage manufacturing process knowledge and proposing a process planning design method, this approach deduced the required processing technology for products based on design model information, thereby fostering process knowledge reuse in manufacturing process planning. Although these studies differ from the field of construction, the development and application of semantic models can offer valuable insights for our research.

In the field of architectural engineering and construction, process knowledge modeling has found diverse applications [13]. By integrating the simple knowledge engineering method and seven-step method, we established a hybrid development approach to create the construction workflow ontology DiCon to facilitate integrated and detailed construction. DiCon was further employed to develop applications for construction process management [14]. In addition, we developed a domain ontology to describe the knowledge model of the multi-stakeholder project development process. Recognizing that the construction process of different projects typically involves numerous repeated but unequal sub-processes [15], we employed an ontology-based process modeling method to support the generation of process plans for construction projects. To achieve safety risk management in prefabricated building construction [16], we constructed an ontology model to represent the construction activity information of the project.

Upon reviewing these studies, we observed a prominent limitation: the majority of workflow modeling has focused on the process information of the entire project, rendering it unsuitable for describing the construction process of building components. In addition, although certain studies have depicted objects similar to those in our investigation, their model applications can primarily serve as a foundation for other work, rather than facilitating information integration and reuse.

Therefore, this study adopted the method of constructing an ontology model to focus on the construction process of fundamental building components or structures. By centering on construction activities as the primary core concept and conceptualizing relevant elements, we achieved the integration and representation of the construction process for specific building components and related information.

### 2.3. Normative Knowledge Modeling and Application

Over the years, the automated compliance inspection of buildings has garnered significant attention from the industry. The ontology approach is valuable for this task, enabling the representation of logical knowledge from electronic regulations or practical experience in a formal and computer-interpretable manner. Compared with hard-coding rules in a computer programming language, this approach offers enhanced readability and changeability [17]. By leveraging ontology and semantic web technology, we established a multi-level knowledge model to represent the concepts and logic of building codes. Similarly, we modeled the relevant knowledge of building codes [18] to focus on realizing an automated energy-performance evaluation of architectural design schemes. Other studies [19] have explored collecting demand information from all parties involved in industrial building production for the automatic inspection of BIM models. Moreover, some researchers automatically infer the next working step based on working conditions [20]. By utilizing the CQIE ontology, the corresponding regulatory constraints are modeled to facilitate the automatic compliance inspection of construction quality during the construction process [21].

However, research on inferring the structural requirements of building components, determining the control parameters in the construction process, identifying the necessary construction activities, and other relevant information based on specifications and empirical knowledge remains limited.

## 3. Methodology

We presented a comprehensive framework for modeling and applying process information related to building components, based on ontology. The framework comprised an ontology layer and a semantic reasoning layer.

### 3.1. Ontology Layer

The ontology layer defined essential concepts and their interrelationships with the field, enabling the integrated representation of the construction process knowledge. Ontology was implemented using machine-readable language, promoting knowledge sharing and reuse.

Due to the vast number of building component categories in the construction field, multiple subcategories may exist under the same type, distinguished by various features. To enable a more convenient representation of the construction technology for diverse building components, we proposed a three-level construction process ontology, including the top-level ontology, generic ontology and specific ontology. The top-level ontology provided essential concepts for describing construction processes and their relationships, serving as a common semantic framework. The generic ontology described common knowledge regarding the construction process information for a specific type of building component. Compared with generic ontologies, specific ontologies served as subtypes, focusing on application objects and offering richer and more detailed knowledge descriptions. Instan-

tiating a specific ontology can represent the construction process of a particular building component. The three levels of ontology were sequentially inherited.

### 3.1.1. Ontology Development Method

Various academic organizations and research institutes have proposed diverse approaches to ontology development tailored to their specific requirements. Upon reviewing popular development methods, notable ones include the IDEF5 Ontology Description Capture Method [22], the TOVE method [23], Methontology [24], and the seven-step method [25]. Methontology primarily focuses on constructing and managing ontology at the knowledge level but lacks a detailed description. The seven-step method emphasizes modeling operational details while lacking a comprehensive and systematic evaluation mechanism and documentation function. In this development endeavor, we amalgamated the strengths of the Methontology method and the seven-step method to propose an improved seven-step method (Figure 1) for developing the construction process information ontology model of building components. The main steps are as follows:

1.  Define the application field and scope of ontology, encompassing the determination of its applicable field, development purpose, and selection of the intended ontology users.
2.  Capture domain knowledge, involving the collection, filtering, and organization of domain knowledge for ontology applications.
3.  Examine the reusability of the existing ontology by searching for relevant reference ontology resources and evaluating their reusability for reuse.
4.  Build the ontology that encompasses abstract concepts in related fields, the definition of hierarchical relationships between concepts, specification of concept attributes and constraints, and instantiation. This step marks the completion of the design and implementation phases of ontology development.
5.  Perform a consistency test of the ontology, identifying potential knowledge gaps and unclear conceptual relationships. If any issues are identified, the ontology development should be revised and returned to the building ontology stage.

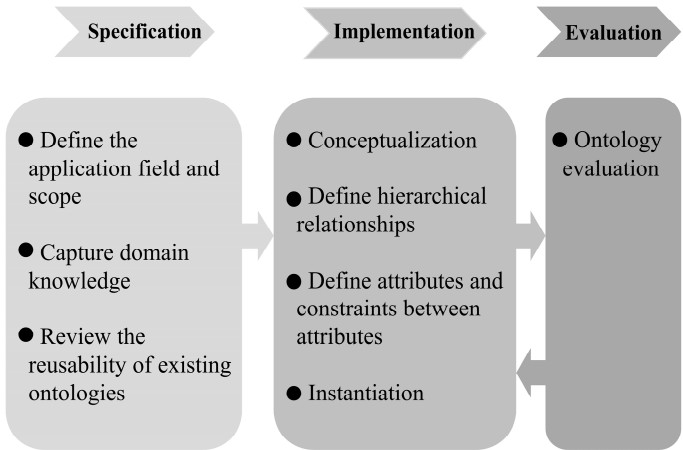

**Figure 1.** Schematic diagram of ontology development method.

### 3.1.2. Core Concepts and Knowledge Sources

Following the ontology development method, the first step involved clarifying the application field of the ontology. In this study, the construction process information ontology was specifically applied in the field of building construction, with a primary focus on integrating the knowledge related to construction processes for various types of building components.

During the related work introduction stage, current process-related ontologies were extensively studied in the fields of mechanical manufacturing and processing. In the construction domain, the majority of the process ontology research centered on the knowledge

management of the construction process of the entire project or the application of ontologies for automated checking tasks, such as safety checking, quality checking, and model compliance review. Although the aforementioned ontologies could provide concepts and attributes for certain building objects, their direct reuse was not feasible due to differences in fields, necessitating their redevelopment.

Based on crucial information concerning the construction process of building components, and drawing from the core elements proposed by quality management experts (i.e., human, machine, material, method and environment), we extracted the core concepts for the top-level ontology. These concepts encompassed domain terminology, standard, construction materials, construction tools, construction process, construction products, product components and key measures (Figure 2). Domain terminology originated from specifications serving to elucidate and facilitate the understanding of special terms used in construction materials, tools, product components and the construction process. Construction materials and tools represented the main resources required for the construction process, while the standard provided constraints for other parts of the information. The construction process class embodied the representation of the construction work process. Acting as supplementary descriptions during construction, the key measure class served to expound the operational requirements of construction activities in detail, as well as the demands for materials, tools and other resources. Construction products and product component classes were used to describe the construction products. Specifically, the construction products referred to the building components mentioned earlier, with this distinction made to avoid confusion with the components that comprised the building components. For convenience, the subsequent expression would often replace building components with building products.

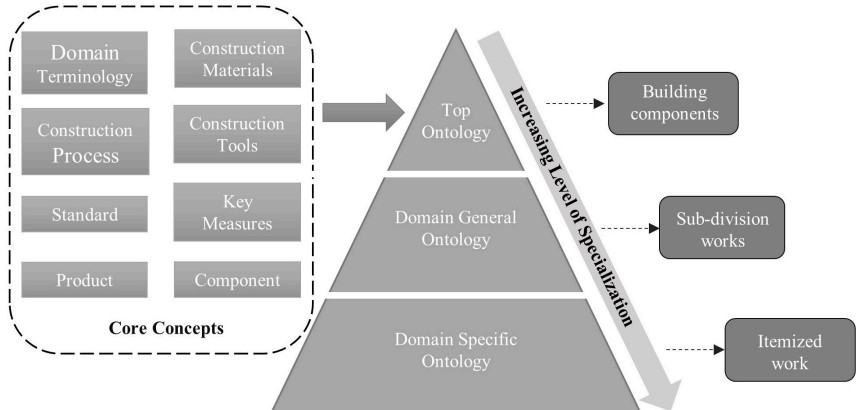

**Figure 2.** Multi-level ontology.

The Unified Standard for Constructional Quality Acceptance of building engineering (GB 50300-2013 [26]) includes a division of part and sub-item projects in Appendix B. We utilized the engineering knowledge of each sub-division project as the knowledge source for each general ontology of each domain, with the sub-item knowledge of the sub-division project employed as the knowledge source for domain-specific ontologies. Sub-items under the same sub-division project typically share the same terminology and standards, while they differ in construction objects, resource requirements, key measures, and construction. Therefore, the domain-specific ontology was built based on the inheritance of the general ontology of the domain. This approach reduced the repeated addition of general knowledge while further refining the information contained in the domain's general ontology to form the construction process plan template.

### 3.1.3. Relation Definition

The relationships within the construction process information ontology were precisely defined, and the naming method was inspired by previous process ontologies. This

included examples such as the digital construction ontology [13] and the ontology method employed to define the construction technology plan [27].

The main concept of the construction process information ontology is the construction process, which comprises a series of activities representing the construction process and its corresponding steps in the process file. The semantic relationships' nomenclature accurately reflects their meaning, facilitating descriptions of the relationships between construction activities and other significant concepts, such as required construction resources, target building components and inter-process relationships. For example, "hastool" and "hasmaterial" delineate the relationship between the construction process of products and the construction tools and materials employed. The "nextprocedure" relationship indicates the time sequence of the process flow, signifying that the implementation of the subsequent construction process should follow the completion of the current process. In addition, defining relationships between elements within the ontology serves as a vital foundation for semantic reasoning and query functions.

The main concepts and inter-conceptual relationships are represented as depicted in Figure 3, collectively constituting the core semantic framework of the top-level ontology. This framework facilitated the expression of construction processes for typical building components within its structure.

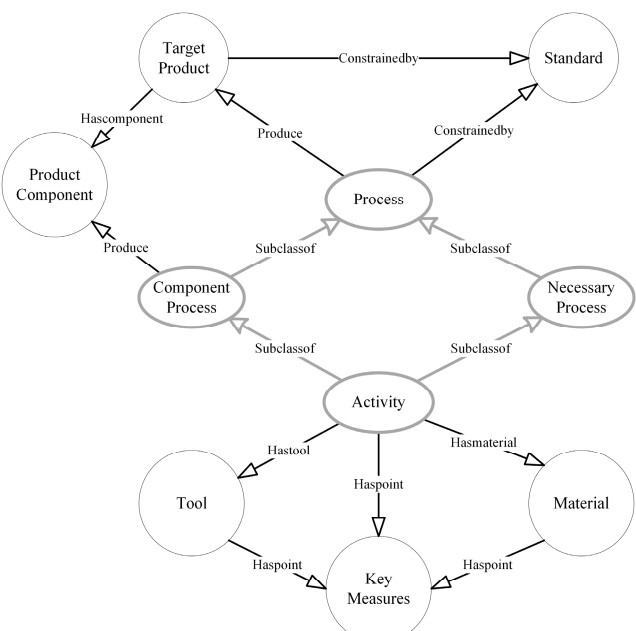

**Figure 3.** Construction process knowledge association model.

### 3.2. Semantic Reasoning Layer

The semantic reasoning layer served the purpose of representing validation, constraint, and query rules in a computer-interpretable language. This facilitated the provision of reasoning and querying functions for the ontology.

To improve the overall quality of the building, both the building component itself and its construction process should adhere to a set of interconnected constraints derived from building codes, expert experiences, engineering practices and other relevant sources. These constraints encompass various types, including, but not limited to, the following:

- Construction requirements of the code for building components with specific geometrical characteristics;
- Parameters requiring attention in construction activities under specific conditions;
- Constraints on the sequence of construction activities in case of changes in the features of building components.

Storing and expressing these constraint rules in natural language required human interpretation and application. However, using a machine-interpretable language for modeling and storing rules significantly affected the compliance assurance of building components and reasoning of process information.

### 3.3. Coding Construction Process and Constraint Rules

In this section, the methodology is further illustrated with engineering cases.

In this case, the sub-division engineering masonry structure engineering of the main structure in construction engineering was selected as the general ontology knowledge source. Domain knowledge was sourced from relevant standards, the content of the masonry engineering part of the engineering manual, the technical disclosure files of similar building components in the past and the experience summaries from construction site staff. The sub-items of masonry structure encompassed brick masonry, concrete small hollow block masonry, stone masonry, reinforced masonry and infilled wall masonry.

The following is an illustration of the frequently employed infilled wall masonry in masonry structures. Utilizing the general ontology of masonry engineering, the lower-level concept of the construction process class was extended to conceptualize construction process activities. The ontology defined the relationship between concepts and instances, thereby generating the process template for the sub-project to facilitate storage and reuse.

The specific ontology concept class, instances, and their relationships are shown in Figure 4.

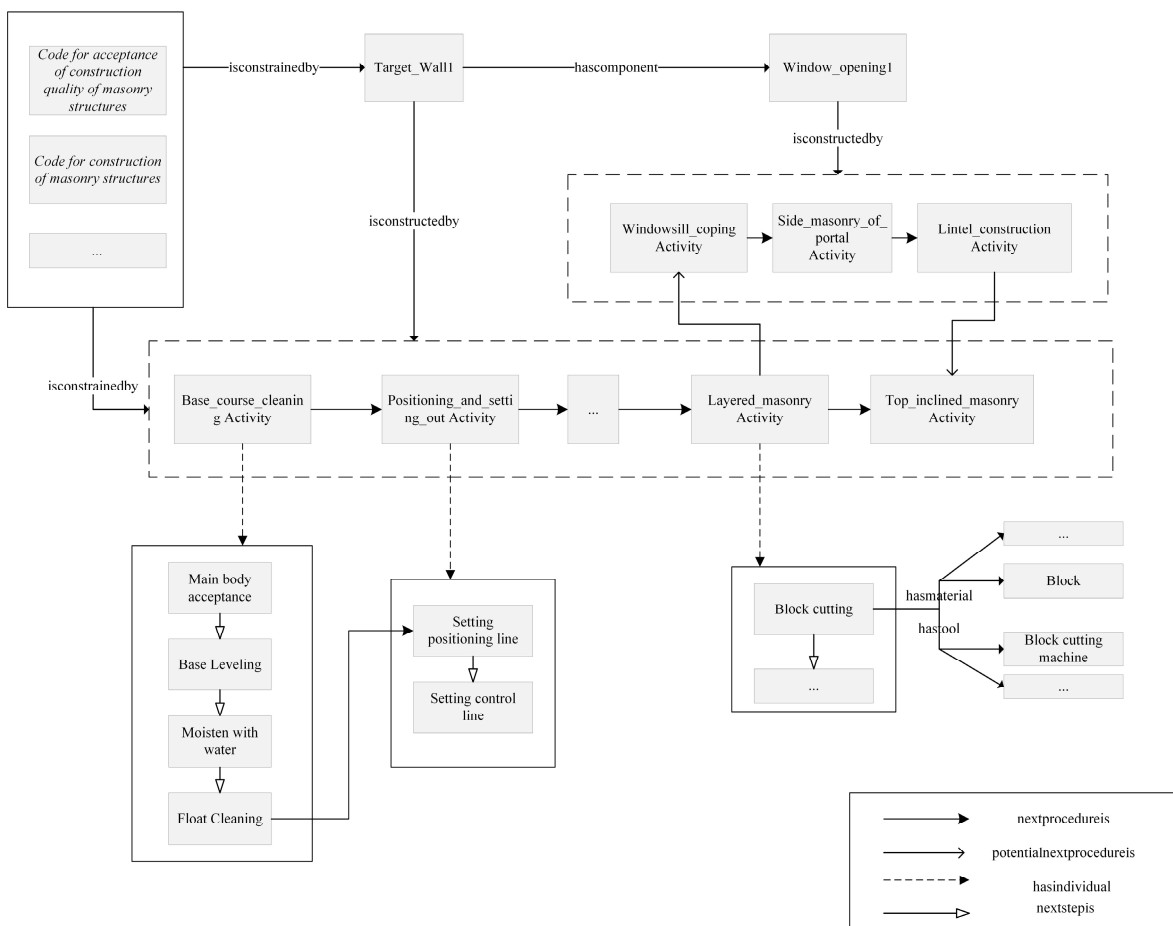

**Figure 4.** Template for infilled wall masonry construction process.

It was noteworthy that the construction process of masonry engineering was dynamic, and that the characteristics of the construction target wall exerted an influence on the

construction process. For example, if the overall size factor of the wall necessitated the addition of other wall components, the construction process would undergo local changes and updates. The relationship "potentialnextprocedureis" was used to represent the potential sequence relationship between construction activities.

3.3.1. Semantic Model Representation

Ontology was established using the OWL language, wherein concepts were represented as classes with a well-defined hierarchical structure. The relationships between conceptual classes and instantiated objects of a class were described using object properties, data properties, and annotation properties. Object properties were utilized to depict the relationships between ontology classes or instances. The data attributes described the feature properties of instantiated objects of classes, and established associations between the specific values and instances of classes. Annotation properties facilitated additional and more comprehensive descriptions of instance objects.

Considering the example of base course cleaning, which constituted the initial step in the construction process of infill wall masonry, process activities were conceptualized as activity classes, with base-course cleaning activities serving as subcategories under the necessary construction process classes for infilled wall masonry. The object property "produce" was used to signify that the role of the construction activity was to build the wall. Moreover, the object property "nextprocedureis" outlined the subsequent procedure of base-course cleaning activities, specifically measurement and setting-out work. This relationship was expressed an OWL axiom that described and constrained the construction process. This axiom operated at the class level, signifying that all positioning and setting-out work activities occurred after the grass-roots cleaning. Furthermore, the "comment" label in the annotation property was utilized to elucidate the activity and enhance its semantic scope. The above description was encoded in the OWL RDF/XML syntax as follows:

```
<owl:Class rdf:about="# Base_course_cleaning Activity">
<rdfs:subClassOf rdf:resource="# Necessary_process" />
<rdfs:subClassOf>
<owl:Restriction>
<owl:onProperty rdf:resource="# produce" />
<owl:someValuesFrom rdf:resource="# Target_wall" />
</owl:Restriction>
</rdfs:subClassOf>
<rdfs:subClassOf>
<owl:Restriction>
<owl:onProperty rdf:resource="# nextprocedureis" />
<owl:allValuesFrom rdf:resource="# Positioning_and_setting_out Activity" />
</owl:Restriction>
</rdfs:subClassOf>
<rdfs:comment>"Refers to the cleaning work carried out before the construction of
the bottom layer of the wall"</rdfs:comment>
</owl:Class>
```

The instantiation of the class signified the representation of process activities incorporated into the work step activities. For instance, considering the base-course cleaning activities, specific examples included the main body acceptance operation, base-leveling operation, floating cleaning operation and moistening with water operation. The following fragment was encoded in OWL RDF/XML syntax.

```
<owl:NamedIndividual rdf:about="# Base_course_cleaning Activity">
<rdf:type rdf:resource="# Base leveling" />
<rdf:type rdf:resource="# Float cleaning" />
<rdf:type rdf:resource="# Main body acceptance" />
<rdf:type rdf:resource="#Moisten with water" />
</owl:NamedIndividual>
```

Object and data properties, similar to mathematical functions, possessed definition and value domains, which were applied to define the body and action object of the property, respectively. The domain and value domain of ontology object properties were primarily derived from class-related content, whereas the value domain of data stemmed from specific values. In defining properties, it was imperative to establish the domain and range to standardize and constrain the ontology, thereby ensuring accuracy in subsequent applications and reasoning. Relationships between properties were defined, with the most notable being reciprocal relationships. Reciprocal relationships referred to object properties with anti-properties, wherein a reciprocal association existed between the properties and their anti-properties. For example, if class A was connected to class B through object properties with anti-attributes, class B would likewise be associated with class A through the anti-properties of the object properties.

The hierarchical relationship of properties, as well as the definition of the application subject and scope, were described in the OWL RDF/XML syntax. Considering the object property "hastool" as an example, the application subject was the construction process, whereas the application scope encompassed the construction tool. Moreover, an anti-property labeled "istoolof" was established.

<owl:ObjectProperty rdf:about="# *hastool*">
<rdfs:subPropertyOf rdf:resource="# *topObjectProperty*" />
<owl:inverseOf rdf:resource="# *istoolof*" />
<rdfs:domain rdf:resource="# *WallbuildingProcess*" />
<rdfs:range rdf:resource="# *Construction_tool*" />
</owl:ObjectProperty>

The data property, exemplified by "Length", was associated with the application subject, which represented the target to build the wall. The application scope further designated the data type as "int".

<owl:DatatypeProperty rdf:about="# *Length*">
<rdfs:subPropertyOfrdf:resource="http://www.w3.org/2002/07/owl#topDataProperty (accessed on 1 August 2023)" />
<rdf:type rdf:resource="http://www.w3.org/2002/07/owl#FunctionalProperty (accessed on 1 August 2023)" />
<rdfs:domain rdf:resource="# *Target_wall*" />
<rdfs:range rdf:resource="http://www.w3.org/2001/XMLSchema#int (accessed on 1 August 2023)" />
</owl:DatatypeProperty>

### 3.3.2. Rule Modeling

This method adopted the SWRL language as the rule modeling language, which evolved from the rule markup language RuleML and OWL ontology. SWRL is a language that presents rules in a semantic manner and provides a high-level abstract syntax that extends OWL semantics [28]. By combining rules and OWL ontology, the SWRL language compensated for the limitations of OWL ontology in reasoning and description. Compared with other reasoning expressions, the SWRL language demonstrated superior logical reasoning and rule expression capabilities, along with improved transitivity, sharing, and reusability. The rules consisted of antecedents (body) and consequents (head). The antecedent represented the premise of rule reasoning, while the consequent embodied the results of rule reasoning. Both the antecedent and consequent were composed of a set of (possibly empty) atoms, including C($x$), P ($x$, $y$), SameAs ($x$, $y$) and DifferentFrom ($x$, $y$), or built-in ($r$, $x$, . . .) expressions. Here, $x$ and $y$ represent variables; C stands for the OWL concept description or data range; P signifies the OWL attribute; and $r$ denotes a built-in relationship. A simple example rule can be expressed as:

$$hasParent(?x1,?x2)\hat{}hasBrother(?x2,?x3)=>hasUncle(?x1,?x3)$$

This rule means that if B is the father of A and B shares a brotherly relationship with C, then C is deemed to be the uncle of A.

Utilizing the construction process information ontology developed above, ontology classes and associated properties served as the foundation for rule formulation. Based on the constraints outlined in Section 3.2, pertinent elements relevant to the construction process information reasoning were identified. Subsequently, SWRL rules were constructed to facilitate the reasoning process concerning building component compliance, process routes and process parameters.

As an illustration, considering the specialized ontology of masonry engineering, the inference rules were derived from the Code for Acceptance of Construction Quality of Masonry Engineering [29] and the engineering manual. These rules were formulated to ensure building component compliance and improve the process information knowledge within the ontology. Some inference rules are presented in Table 1.

**Table 1.** Some normative rules.

| Rule | Antecedent | Consequent |
|---|---|---|
| Rule 1 | There are holes inside the wall, and the width of the hole is greater than 2000 mm | Structural columns need to be set on both sides of the opening |
| Rule 2 | There are holes inside the wall, and the width of the openings is greater than 300 mm | The beam should be set at the top of the hole |
| Rule 3 | The length of the independent wall is greater than 2.5 m | Structural columns should be set at both ends of the wall |
| Rule 4 | The height of the wall is greater than 4 m | A ring beam should be set in the middle of the wall |
| Rule 5 | The mortar is made and the ambient temperature is greater than 30 degrees Celsius | The mortar is used for no more than 2 h |
| Rule 6 | Additives are added to mortar making | The stirring time must exceed 180 s |

The SWRL rule language is employed to depict Rule 1 as follows:

**Rule 1.** *Target_wall(?w) ^ hascomponent(?w, ?op) ^ swrlb:greaterThan(?data, 2000) ^Width (?op,?data)^Opening(?op)^Structural_column(?sc)->setcomponent(?op,?sc)^Column_location(?op, "set on both sides of the opening")*

The predicates in the SWRL rules were primarily derived from the classes defined in the OWL and the properties associated with these classes. Object properties, such as "hascomponent" and "setcomponent", indicated the presence of holes in the target wall, necessitating the placement of construction columns at these holes. Data properties, namely "Width" and "Column _ location", were used to represent the width of the hole and position for setting the construction column, respectively. Additionally, the built-in function "swrlb: greaterThan" facilitated comparison within the SWRL language. Rule 1 was implemented to assess the compliance of the wall structure and, building upon this evaluation, the process was updated accordingly.

In Table 1, Rules 2–4 exhibited resemblance to Rule 1 and were subsequently transformed into SWRL language, as demonstrated in Table 2.

**Table 2.** Expression of rules in SWRL.

| Rule | Expressed by SWRL |
|------|-------------------|
| Rule 2 | *Target_wall(?w) ^ hascomponent(?w, ?op) ^ swrlb:greaterThan(?op_width, 300) ^ Width(?op, ?op_width ^ Opening(?op) Lintel(?l)-> setcomponent(?op, ?l)* |
| Rule 3 | *Target_wall(?w) ^ Length(?w, ?w_length) ^ swrlb:greaterThan(?w_length, 2500) ^ Link_object(?a, "none")^Structural_column(?sc)-> setcomponent(?w, ?sc) ^ Column_location(?a, "set at both ends of independent walls")* |
| Rule 4 | *Target_wall(?w) ^ Height(?w, ?w_height) ^ swrlb:greaterThan(?w_height, 4000) ^ Ring_beam(?rb)-> setcomponent(?w, ?rb) ^ Ring_beam_location(?w, "set at the middle and high part of the wall")* |

In relative terms, the primary function of Rules 5 and 6 was to govern the construction operation activities and articulate the operational aspects of construction activities under specific conditions, aiming to minimize potential quality issues with the wall. These rules can be represented in SWRL as follows:

**Rule 5.** *Mortar(?m) ^ Operation_temperature(?m, ?ot) ^ swrlb:greaterThan(?ot, 30) -> pos:Shelf_life (?a, "<=2h")*

**Rule 6.** *Making_mortar(?mm) ^ hasmaterial(?mm, ?ma) ^ Mortar_Additive(?ma) -> Operation_time(?mm, ?ot) ^ swrlb:lessThanOrEqual(?ot,180)*

The aforementioned rules served as illustrations for the automatic inference of additional wall components and considerations in construction activities, based on the features of the target wall and the relevant construction information. These rules were derived from normative provisions and played a pivotal role in ensuring compliance with the entire wall construction process.

To achieve the reuse of workflow, based on some of the above specification-based rules, Rules 7 and 8 were modeled as follows:

**Rule 7.** *Target_wall(?w) ^ Necessary_process(?np)->hasprocess(?w,?np)*

**Rule 8.** *Target_wall(?w) ^ Structural_column(?sc) ^ setcomponent(?w, ?sc) ^ Structural_column-Formwork_removal(?scfr) ^ Structural_colunm-Concrete_pouring(?sccp) ^ Structural_column-Installation_of_formwork(?sciof)^Structural_column-Steel_bar_binding(?scsbb)->hasprocess(?w,?scfr) ^ hasprocess(?w,?sccp) ^ hasprocess(?w,?sciof) ^ hasprocess(?w,?scsbb)*

The initial example rule aimed to automatically deduce the necessary process for the target building wall. The subsequent rule represented second-level reasoning for the required process, based on the premise that the required wall components were inferred using the aforementioned rules. The latter rule specifically implied that, in instances where the wall required the incorporation of structural columns, it may become essential to include the construction processes for structural column steel bar binding, structural column concrete pouring and other relevant construction activities.

## 4. Implementation and Validation

### 4.1. Ontology Implementation

We employed Protégé 5.5.0, developed by Stanford University, as the platform for implementing the ontology and validating its application. Protégé is an open-source software tool that offers a significant advantage over other ontology editing software by supporting Chinese representations, thereby enhancing the applicability of this research. In Protégé, the entire infilled masonry process ontology was visually depicted, encompassing conceptual classes, defined relationships, associated attributes and constraints.

Using Protégé as the basis, object properties can be further defined to encompass characteristics such as Functional, Inverse Functional, Transitive, Symmetric and Asymmetric. The primary classes and their attributes within the masonry construction process information ontology were implemented in Protégé, as illustrated in Figure 5.

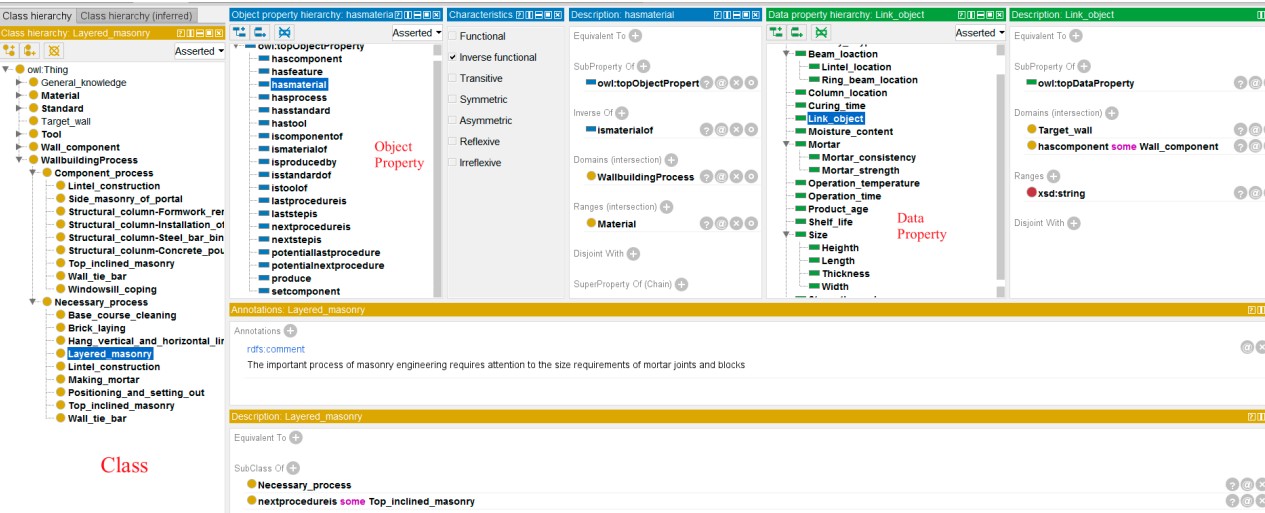

**Figure 5.** Ontology constructed in Protégé 5.5.0.

The instantiation of ontology classes represents the concrete embodiment of the concepts described within the ontology. This involves the further expansion and refinement of the ontology after completing the class and property definitions. In Protégé, instantiation can be achieved using the built-in Cellfie module, which facilitates the rapid import of information. As an illustration, Figure 6 demonstrates the instantiation of a construction activity class in Protégé.

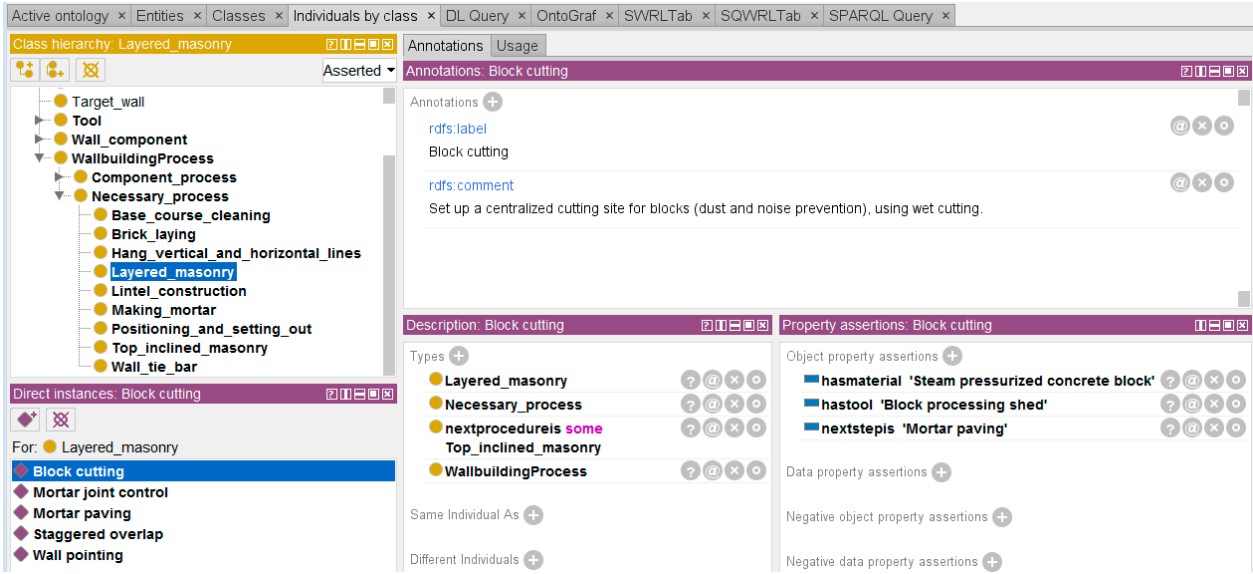

**Figure 6.** Instantiation of ontology classes in Protégé 5.5.0.

## 4.2. Consistency Check

Once the construction process information ontology was finalized, it was imperative to conduct an evaluation and quality check to ensure compliance with the construction principles and usage standards. This step was vital to guarantee the consistency of knowledge representation and the accuracy of ontology reasoning.

The inference engine adopted in this study was Pellet, developed by the University of Maryland in the United States. Pellet exhibits Java interface compatibility and offers a comprehensive range of standard inference services typically provided by deep learning inferencers. These services include consistency checking and determination of subclass relationships.

Building on the previously proposed ontology construction method and approach, we utilized Protégé and the inference engine to conduct a thorough consistency evaluation of the process information ontology. The evaluation covered syntax, semantics, user customization and other relevant aspects. Based on the evaluation results, we made necessary adjustments to the hierarchical structure, attribute constraints and related instances of the ontology to mitigate potential internal conflicts during its practical usage. In cases where ontology classes exhibited inconsistencies, such conflicts were displayed and highlighted in red, along with corresponding explanations.

As shown in Figure 7, the inference function was enabled. There were no red marks on the platform, indicating that this ontology did not exhibit any inconsistencies in the definition of relationships between classes and instances.

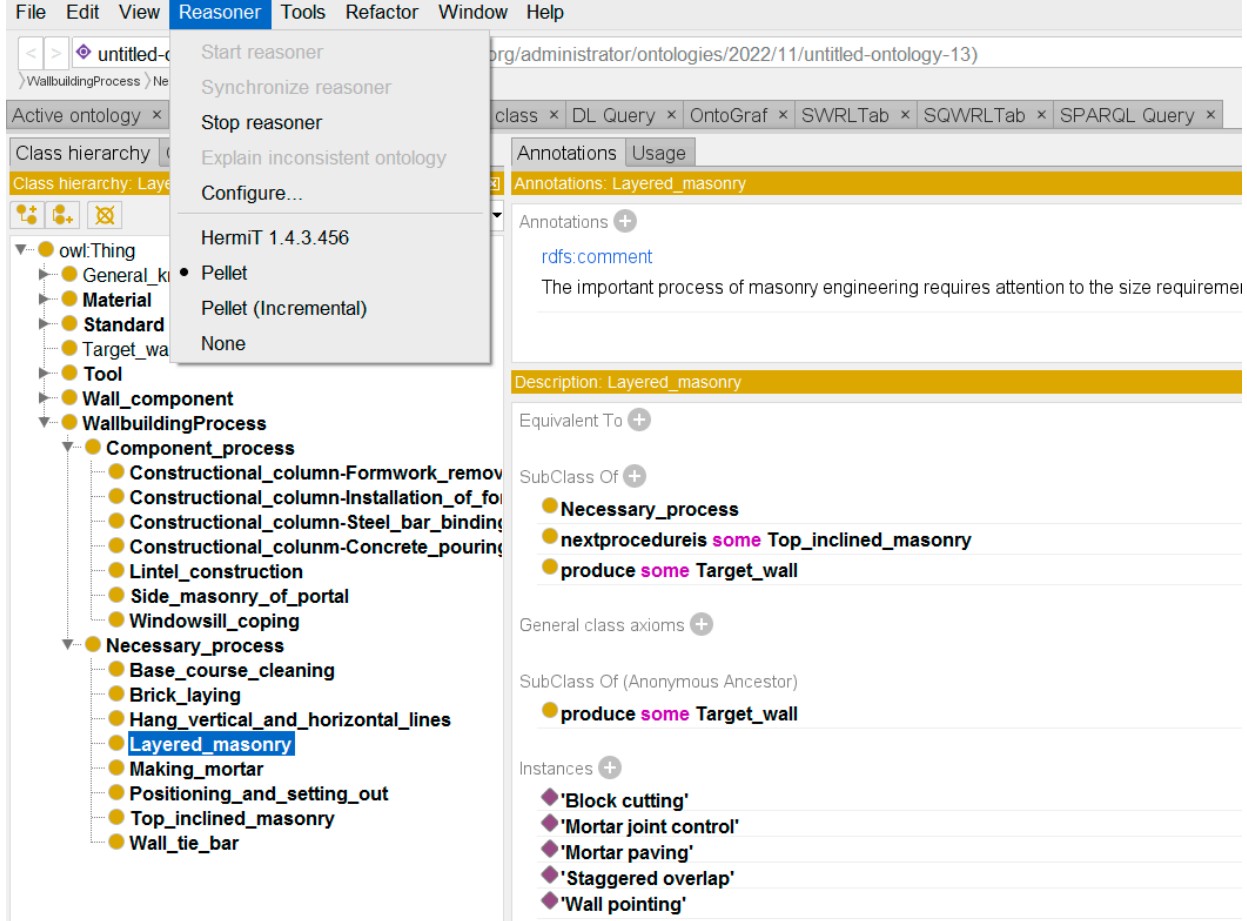

**Figure 7.** Results of consistency testing.

*4.3. Rule-Based Reasoning*

Upon modeling the constraint rules in SWRL, we utilized the rule-reasoning engine Drools, which was based on the Java language, to perform knowledge reasoning related to process information. The reasoning process, depicted in Figure 8, is as follows. Firstly, the constructed SWRL rules were imported into the inference engine. Subsequently, the ontology knowledge and inference rules were transformed based on the Drools engine. The transformed content was then matched and reasoned using the Drools inference engine algorithm. Finally, the results of the Drools inference were transformed into OWL knowledge and displayed in the ontology implementation tool Protégé.

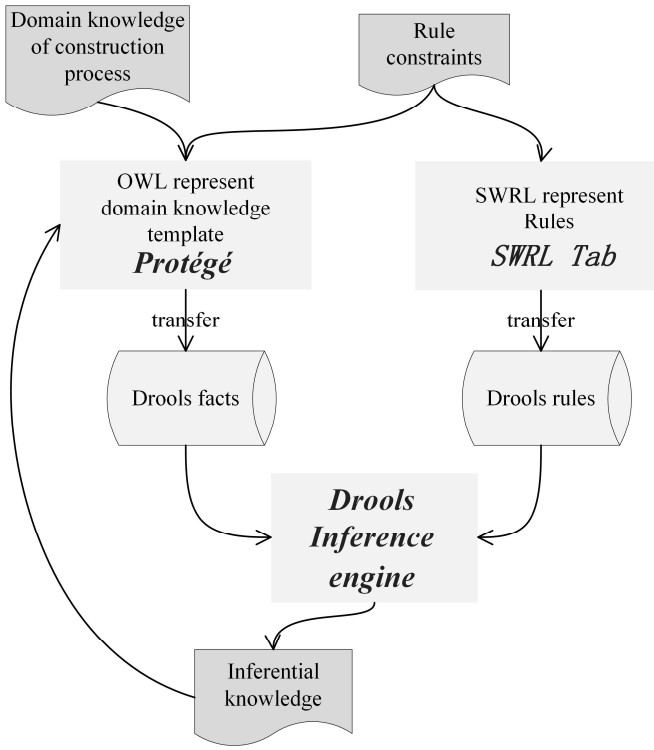

**Figure 8.** System for knowledge reasoning.

Protégé is equipped with its own Drools inference engine, which facilitates the import of SWRL rules and execution of Drools rule inference via the software's SWRLTab plug-in. In combination with the previously developed construction process information ontology for infilled walls, and the constructed SWRL rule example mentioned in Section 3.3.2, we performed reasoning. By inputting the wall instance and executing Rule 3, as mentioned in Section 3.3.2, we obtained the inference results shown in Figure 9.

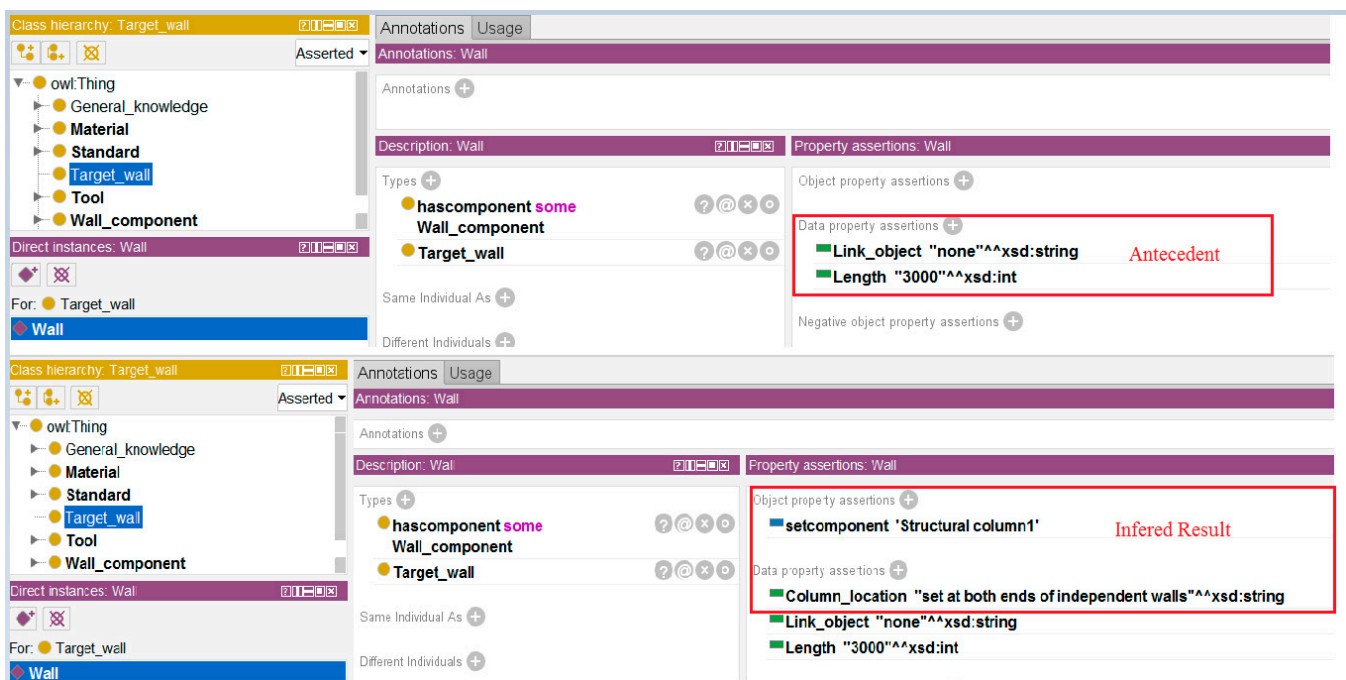

**Figure 9.** Screenshots of the reasoning results of the required wall components.

The obtained reasoning results demonstrated that, to ensure the compliance of the wall structure, it was necessary to set and position the structural column on both sides of the wall.

In the context of the aforementioned wall-feature-based reasoning, Rules 7 and 8, as discussed at the end of Section 3.3.2, were executed to automatically deduce the construction activities required for the wall construction.

The successful inference of the construction activities required for the wall is illustrated in Figure 10. The sequence of all the construction activities was defined within the ontology, Construction activities were also related to relevant information through attributes. The results revealed that this method can improve information relevance and enable the efficient reuse of process knowledge.

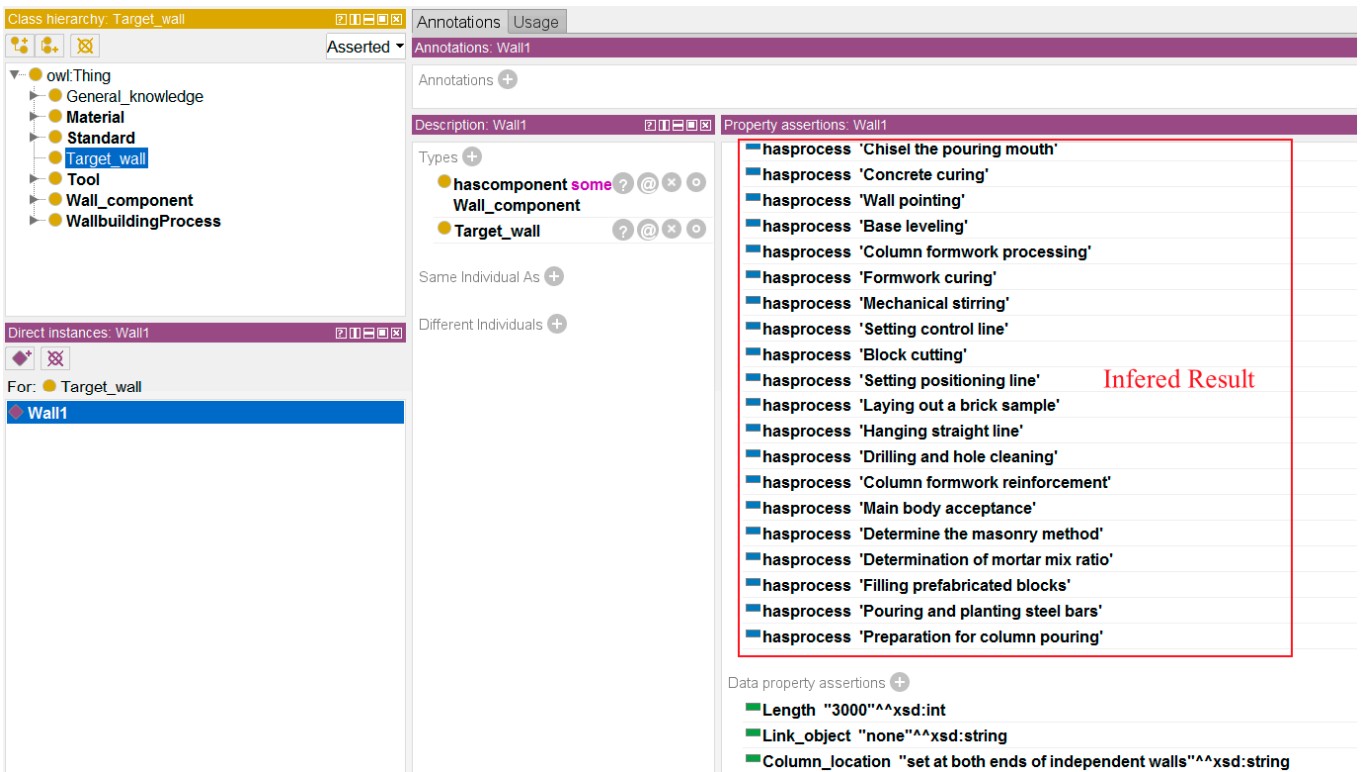

**Figure 10.** Screenshots of the required construction activity reasoning results.

### 4.4. Query Functions

The construction process information ontology of building products can be intricate and the volume of knowledge is substantial. A mere visual query may not adequately capture process knowledge. Therefore, this study constructed a knowledge query mechanism using the ontology query language SPARQL, which enabled users to efficiently retrieve construction process information, enhance their understanding of construction process particulars and mitigate construction quality issues.

SPARQL is a query language and data-acquisition protocol designed for RDF development. By employing SPARQL, users can construct queries to extract specific resource information adhering to the W3C RDF specification [30]. This language can be highly valuable for efficiently retrieving necessary information from extensive semantic web data.

Compared with the aforementioned ontology rule reasoning and its application functions, the query statement is a practical function for users. Hence, the SPARQL query statement developed in this study was designed based on the actual requirements for knowledge retrieval and querying in the construction process knowledge. It encompassed the pertinent normative provisions, construction parameters, process flow, construction

key points, quality control points, construction tools and construction materials relevant to the construction process and past experiences.

The step sequence knowledge query of the base-course cleaning activity within the construction process information ontology for infilled wall masonry is as follows:

1. ***Select*** *?step ?step2*
2. ***where*** *{ ?step a pos:Base_course_cleaning.*
3. *?step bas:nextstepis ?step2.}*

Executing the query statement yields the result shown in Figure 11.

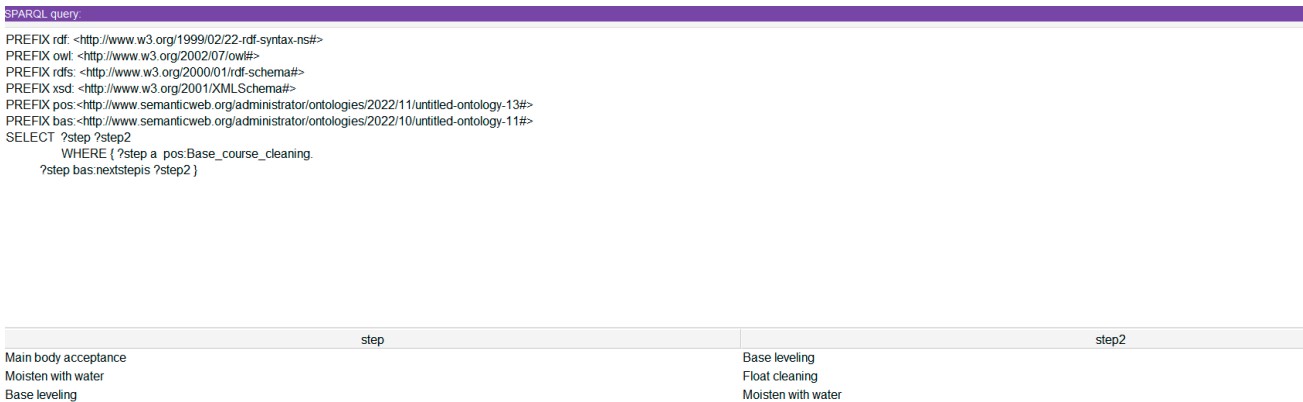

**Figure 11.** Step sequence query of base-course clean-up activities.

In the ontology, the "comment" label can be used to express the operational guidelines and quality control criteria for each construction step of layered masonry work, thereby facilitating subsequent information retrieval:

1. ***Select*** *?step ?comment*
2. ***where*** *{ ?step a pos:Layered_masonry.*
3. *?step rdfs:comment ?comment.}*

Executing the query statement produces the result shown in Figure 12.

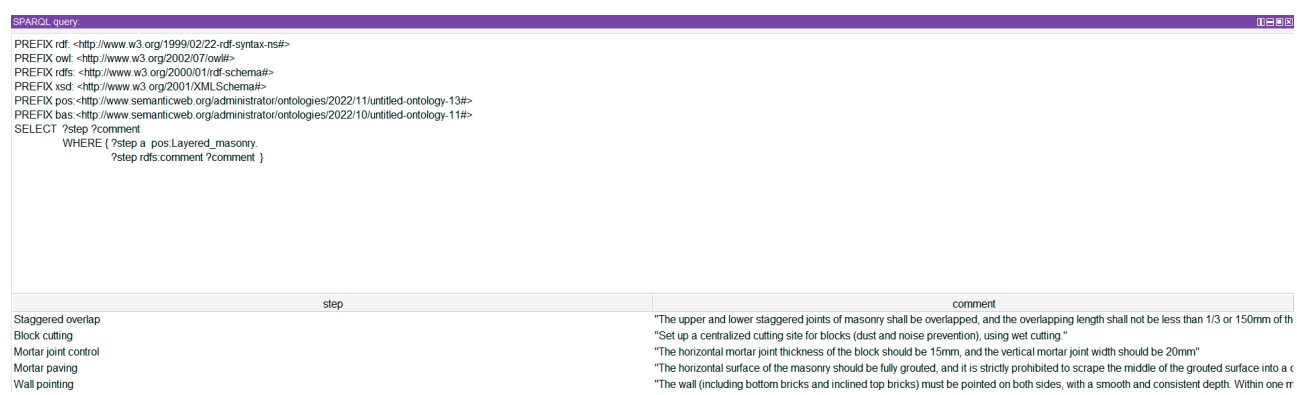

**Figure 12.** Query for construction process operation and quality control points.

A query was conducted for the materials and tools required for the construction process of layered masonry work:

1. ***Select*** *?step ?material ?tool*
2. ***where*** *{ ?step a pos:Layered_masonry.*
3. *?step pos:hasmaterial ?material.*
4. *?step pos:hastool ?tool.}*

Executing the query statement yields the result shown in Figure 13.

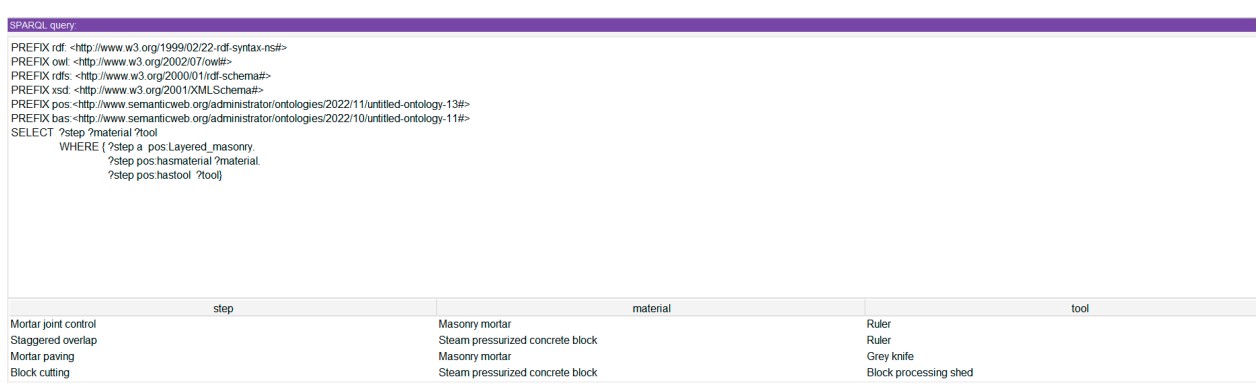

**Figure 13.** Query for required procedures, materials and tools.

## 5. Conclusions

This study proposed an ontology-based approach aimed at enhancing the representation and reuse of construction process information in the construction domain. The method was demonstrated and evaluated using a typical construction process scheme of infilled wall masonry in a masonry structure engineering sub-project, along with pertinent constraint provisions from the building code. The results confirmed that this method facilitated knowledge verification, reasoning and reuse in the construction process. The contributions of this study are as follows:

1. This study introduced innovations in the application domain, as most ontologies in the construction field have primarily targeted the integration of workflow information for entire projects. In contrast, our approach concentrated on integrating typical construction process information of specific building components or structures, such as walls and columns, within the construction domain.

2. The ontology model served as a means to integrate the construction process information, offering several advantages over the discrete information contained in traditional process files. It notably improved the relevance and readability of the information while also serving as an efficient source for technical disclosure.

3. A multi-level ontology was introduced to represent the construction processes of sub-projects within the construction domain. Building on this foundation, the ontology can be readily modified and extended to cater to the specific application objects. As a result, the construction processes of other types of building components can be similarly modeled and subjected to reasoning.

4. SWRL was employed to model the constraint rules for reasoning construction process information related to building components. This approach enhanced the efficiency of reusing the construction process information while ensuring the accuracy and correctness of the information.

As an exploratory approach with experimental characteristics, this method requires further research to address its limitations. For example, the instantiation of ontology classes in the verification process is currently achieved by importing data through the built-in block Cellfie in the Protégé-OWL editor. However, certain existing studies [31] have explored different methods, such as using the developed IFC data model for EXPRESS-to-OWL converters [32], utilizing BIM models to represent building information, exporting it to the IFC format and subsequently transforming it into ifcOWL. This facilitates the automatic formation of ontology instances and related attribute information, thereby providing a direction for rapid ontology instance construction.

Finally, this method enabled the creation of a dedicated ontology that comprehensively subdivided the field, catering to the reuse of construction process knowledge across various sub-divisional projects. Although this step may be laborious during the initial application of the method, it verified the versatility of the method. However, as the accumulation of specialized ontology storage grows, this issue can be alleviated.

**Author Contributions:** Conceptualization, L.J. and Y.J.; methodology, L.J. and Y.J.; software, Y.L. and J.L.; validation, L.J., Y.J., Y.L. and J.L.; formal analysis, L.J.; investigation, Y.J.; resources, L.J.; data curation, Y.J. and J.L.; writing—original draft preparation, L.J., Y.J. and Y.L.; writing—review and editing, L.J., Y.J., Y.L. and J.L.; visualization, Y.J.; supervision, Y.J. and Y.L.; project administration, L.J.; funding acquisition, L.J. All authors have read and agreed to the published version of the manuscript.

**Funding:** This research was funded by the Science and Technology Plan Project of the Jiangxi Geological Bureau (2021JXDZ70001), and the Science and Technology Plan Project of the Jiangxi Coalfield Geological Bureau (2020JXMD70003).

**Data Availability Statement:** Not applicable.

**Acknowledgments:** The authors thank all the personnel who either provided technical support or helped with data collection. We also acknowledge all the reviewers for their useful comments and suggestions.

**Conflicts of Interest:** The authors declare no conflict of interest.

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
