# Peer review of "Ontological Method for the Modeling and Management of Building Component Construction Process Information"

_buildings, doi:10.3390/buildings13082065_

Round 1
Reviewer 1 Report
Article meets the requirements for publication in this journal.
However, the author had to revise as follows:
Abstracts are emphasized with findings that are clearly described, abstracts that are too long, are given a maximum value of 250 words but do not reduce the essence of the findings of this study.
The scale must be included in the attachment and source citation
Reliable data and proper research methods.
Adjust the numbers following the library order hypothesis.
References may only be cited in journal articles or books. Mastering literature such as dissertations, internet materials, and reports. Referenced journal articles must have volume, issue, and page information, DOI.
It is not necessary to write down all the statistics, it is enough to replace them in the description form
References must be written in APA style.
For conclusions, provide results that do not differ in interpretation, statistics do not need to be rewritten
Please fix the writing template according to the template
Pay attention to little things like typos, quote styles, and so on.
For images or graphics that come from copy paste results when processing data, they can be included in the attachment. You replace it in the form of an explanation to shorten the page count of your article.
Interesting research, just needs improvement in accordance with the description I conveyed.
Reviewer 2 Report
REVIEW COMMENTS
I have only a few concerns about the paper and some suggestions that maybe the authors could consider:
1. Initially, it is important to address the presence of typographical errors and grammatical inaccuracies within the text. Furthermore, the inclusion of lengthy sentences may result in reader confusion and should be carefully considered for improved clarity and comprehension.
2. In the 'Introduction' section, the proposed research gap and the stated objectives do not meet the criteria of proper synergy. Please make the research gap and the research objectives consistent with each other.
3. The authors used the abbreviation “SWRL” for the first time without clarifying the full name of this abbreviation. So, the authors should write the full name first time.
4. I recommend enhancing the "Introduction" and “literature review” sections by incorporating more recent references to bolster its quality. Three articles that I propose are "neuromarketing tools used in the marketing mix: a systematic literature and future research agenda", " consumer behaviour to be considered in advertising: a systematic analysis and future agenda", and "exploring factors influencing neuromarketing implementation in malaysian universities: barriers and enablers". These references have the potential to enrich your introduction by introducing captivating topics and engaging insights.
5. I suggest the organizational structure of the papers used section 1, section 2, and so forth to ease know the content of the paper.
6. The authors should clarify the advantages of Protégé 5.5.0 software over other software such as but not limited NVivo.
7. The authors should explicitly state the novel contribution of this work and its similarities and differences with their previous publications.
8. The authors need to clearly articulate the theoretical as well as practical implications of this study, which can be named the theoretical and practical implications of this study. I suggest ref. "exploring global trends and future directions in advertising research: a focus on consumer behavior", which can be useful for this issue.
9. The authors need to clearly articulate the future research of this study in the limitations section.
10. For readers to quickly catch your contributions, it would be better to highlight major difficulties and challenges and your original achievements to overcome them in a clearer way in the abstract and introduction.
11. How could/should your study help future studies?
12. The authors should be more careful about references if there are any duplicated references.
If these revisions can be made to the manuscript, I believe that this study can be accepted for publication.
I wish the authors all the very best with this study.
Initially, it is important to address the presence of typographical errors and grammatical inaccuracies within the text. Furthermore, the inclusion of lengthy sentences may result in reader confusion and should be carefully considered for improved clarity and comprehension.
Reviewer 3 Report
It is difficult for me to discern the scientific contribution of this paper. The paper is very poor. In the opinion of the reviewer, the paper needs a serious reworking before it will be ready to be published in an international journal.
The introduction is not clear. The methodology is confusing. No hypothesis has been set, nor are the goals and sub-goals of the paper clearly defined. The results and discussion are poor. Unfortunately, there are several insufficiencies that need to be improved.
Here are comments for the Authors for their further improvement:
1. The introduction (Section 1) and the section 2 (related work?) describe the research background very poor. The paper does not present the full background of the issue under study.
2. I do not see the aim of this research (is not clear and did not present in the paper).
3. The next Section (Knowledge modeling) should be dedicated to describing methodology and what Author did in paper. The methodology should be described and be solid enough such that any other person using the same procedure will could repeat the research. Now, it is impossible because paper do not have this section.
4. The manuscript is insufficient structured with no real stated connection between the sections. The authors need to rearrange their sections and use the IMRaD structure.
5. In this paper I do not see good prepare section: Discussion. Unfortunately, this paper does not point out the shortcomings of past research to show the value of this research. Add a strengths and weaknesses section and limitations section of this research to the new section: Discussion. The discussion should refer to other studies, indicate the shortcomings of the research.
The final impression is that it is necessary to make a complete restructuring of the paper, strengthen the research, and then present it in a quality manner. In this form, unfortunately, I suggest rejecting the paper.
Round 2
Reviewer 2 Report
All of my concerns have been fully addressed in the revised version.
Good luck to the authors in the future publications
Initially, it is important to address the presence of typographical errors and grammatical inaccuracies within the text. Furthermore, the inclusion of lengthy sentences may result in reader confusion and should be carefully considered for improved clarity and comprehension.
Reviewer 3 Report
Thank you to the authors for improving the paper.
All comments have been taken into account in the submitted version of the article